# Epidemiology of Cancers in Men Who Have Sex with Men (MSM): A Protocol for Umbrella Review of Systematic Reviews

**DOI:** 10.3390/ijerph17144954

**Published:** 2020-07-09

**Authors:** Manoj Kumar Honaryar, Yelena Tarasenko, Maribel Almonte, Vitaly Smelov

**Affiliations:** 1Prevention and Implementation Group (PRI), International Agency for Research on Cancer (IARC), World Health Organization (WHO), 150 Cours Albert Thomas, 69372 Lyon, France; mkhonaryar@gmail.com (M.K.H.); almontem@iarc.fr (M.A.); 2Service des Urgences, Hôpital Lariboisiere, Assistance Publique Hôpitaux de Paris (Aphp), 2 Rue Ambroise Paré, 75010 Paris, France; 3Jiann-Ping Hsu College of Public Health, Georgia Southern University, 501 Forest Dr, Statesboro, GA 30458, USA; ytarasenko@georgiasouthern.edu; 4Division of Noncommunicable Diseases and Promoting Health through the Life-Course, WHO Regional Office for Europe, UN City, Marmorvej 51, 2100 Copenhagen, Denmark

**Keywords:** men who have sex with men, cancer, epidemiology, umbrella review

## Abstract

While earlier studies on men having sex with men (MSM) tended to examine infection-related cancers, an increasing number of studies have been focusing on effects of sexual orientation on other cancers and social and cultural causes for cancer disparities. As a type of tertiary research, this umbrella review (UR) aims to synthesize findings from existing review studies on the effects of sexual orientation on cancer. Relevant peer-reviewed systematic reviews (SRs) will be identified without date or language restrictions using MEDLINE, Cochrane Database of Systematic Reviews, and the International Prospective Register for Systematic Reviews, among others. The research team members will prepare the data extraction forms. Two reviewers will independently assess extracted SRs using the Assessment of Methodological Quality of Systematic Reviews. A third reviewer will weigh in to resolve discrepancies. The reviewers will be blinded to publisher, journal, and authors, making their judgements on the title, year, and abstract. The Preferred Reporting Items for Systematic Reviews and Meta-analysis checklist will guide data synthesis. By collating evidence from multiple reviews into one accessible and usable document, our first UR on global epidemiology of malignancies among MSM would serve as an evidence-based decision-making tool for the public health community.

## 1. Introduction

Cancer is a major cause of morbidity and mortality worldwide, with an estimated 18.1 million new cases and 9.6 million deaths in 2018. If recent trends in cancer incidence and population growth continue globally, there may be 27.5 million new cancer cases globally each year by 2040. This estimate is an increase of 61.7% from 2018. The increase is expected to be higher in males (67.6% increase) than in females (55.3% increase) [1].

The term “men who have sex with men” (MSM), according to the Joint United Nations Programme on HIV and AIDS (UNAIDS) Action Framework [2], is used to describe males who have sex with other males, regardless of whether or not they have sex with women or have a personal or social identity associated with that behavior, such as being “gay” or “bisexual”. The term “MSM” has been used in epidemiologic literature since 1990s as an identity-free term to avoid complexities of social and cultural contexts. Focus on increased risk of acquiring sexually transmitted diseases (STDs), including Human Immunodeficiency Virus (HIV), in MSM compared to women might have shifted the scientific use of the “MSM” term from identity-based to behavior-based [3].

Worldwide, estimates of the MSM prevalence and their health behaviors are highly limited for a number of reasons, such as challenges to facilitating discussion of sexual behaviors in current medical practice [4] and population-based surveillance [3]. Data availability varies between regions and is mostly restricted to countries where MSM populations are visible (e.g., 1.5% in the Netherlands [5], 1.6% in Australia [6], 2.2% in the United States [7], 5.5% in the United Kingdom [8], and 7.8% in Brazil [9]). In low- and middle-income countries (LMIC), the prevalence of MSM at least once in their life ranges between 6–20% [10]. Importantly, these percentages likely underestimate the true MSM prevalence, as methods and instruments of data collection differ across regions. Furthermore, surveying MSM can be challenging if not impossible in countries where they are less visible or stay hidden due to social stigma or religion-based discrimination against sexual-minority individuals [11].

While earlier studies on MSM tended to examine infection-related cancers, an increasing number of studies have emphasized the importance of focusing on effects of sexual orientation on other cancers. The prevalence of several cancer risk factors is higher in this population than men who do not have sex with men. For example, MSM are twice as likely to be current smokers [12] and have frequent eating disorders [13] or high alcohol consumption [14]. Commonly reported intravenous (IV) drug use among MSM [15] increases their risk of acquiring the human immunodeficiency virus (HIV), Hepatitis C Virus (HCV), and other cancer-associated infections [16]. Both sexually transmitted infections (STIs) and non-STIs in MSM are of particular concern because MSM are at a higher risk of viral and bacterial STIs [17], being disproportionately affected by HIV globally [18]. The disproportionate burden can be attributed to the high number of sex partners [19]. In addition, anal sex with exposure of highly susceptible tissues from the anogenital area (i.e., anorectal mucosa, the inner surface of the foreskin, and the urethral meatus [20]) to pathogen transmission is among biological reasons for increased risk of acquiring cancer-causing STIs (e.g., high-risk HPV) among MSM. Furthermore, unlike their heterosexual counterparts, MSM face challenges with health care access, such as the lack of health insurance through the spouse’s employer as same-sex marriages are currently not legalized in many countries [21]. Hence, MSM represent a specific group of interest for cancer prevention and control.

Since several systematic reviews (SRs) on cancer epidemiology among MSM have been published [22,23,24], the goal of our research is to conduct an umbrella review (UR). While there are several definitions of umbrella reviews, their aim is to summarize available evidence, assess similarities and differences in published reviews, and perform different types of synthesis [25]. Hence, our UR will synthesize aspects of the current state of cancer surveillance on sexual orientation and knowledge on this unique population, including cancer risk factors, incidence, mortality, and survival compared to the non-MSM (i.e., population not included in the MSM definition). By doing so, the UR review will make substantive and methodological contributions to global epidemiology of malignancies among MSM. The purpose of this publication is to describe an UR protocol, which will guide our study and ensure its reproducibility.

## 2. Materials and Methods

### 2.1. Study Design

To increase transparency, this study protocol has been registered with PROSPERO databases for systematic reviews and meta-analyses, a web-based international registry of protocols [CRD42017073377] of York University and National Institute for Health Research (NIH) (the United Kingdom). The protocol has been designed and reported according to the reporting guidelines of Preferred Reporting Items for Systematic Reviews and Meta-Analyses Protocols (PRISMA-P) checklist (Annex 1) [26]. We anticipate completing the UR in fall 2020.

As a type of tertiary research, an UR is a systematic collection and assessment of several systematic reviews done on a specific research topic or question [25,27]. The scope for the presented UR will be formulated with the epidemiological indicators, namely incidence (i.e., number of new cases) and prevalence (i.e., number of existing cases) of cancer, as well as survival (e.g., 1-year and 5 -year) and mortality from cancer. All cancer and specific types, as defined by the International Classification of Diseases codes (e.g., 140-209 ICD-9 codes [28] and C00-C97 ICD-10 codes [29]) will be examined. Due to their burden in male population [1], particular cancers and their ICD-O-3 codes are as follows: prostate cancer [C61.9], anal cancer [C21.0] [C21.1] [C21.2] [C21.8], colorectal cancer [C18-20], penile cancer [C60.0] [C60.1] [C60.1] [C60.8] [C60.9], lung cancer [C34.0] [C34.1] [C34.2] [C34.3] [C34.8] [C34.9], and liver cancer [C22.0] [C22.1]. An attempt will be done to stratify the results by HIV status, if available.

### 2.2. Study Population and Studies for Inclusion

The UNAIDS definition will be used to identify eligible for inclusion SRs on cancer among the MSM population (inclusive of all ages, ethnicity, socio-economic status, country of residence or settlement, and HIV-positive or negative status). The definition refers to men who exclusively identify themselves as gays or those as also having sex with women [2]. Only studies that have used a systematic process to search the literature and synthesize the data with standard reporting procedures (e.g., PRISMA) will be eligible for inclusion in the UR. It will not include abstracts, critical reviews, integrative reviews, primary studies, withdrawn/retracted publications, dissertations, and literature reviews.

### 2.3. Outcome Measures

The outcomes of this study will be overall and cancer-specific incidence, prevalence, survival, and mortality rates, as well as established and potential risk factors of all cancers and aforementioned types. More details on definitions of indicators are provided in Table 1.

The UR will follow the Participants, Exposure, Comparison and Outcome (PECO) framework for the risk factors (Table 2).

### 2.4. Search Strategy

Relevant SRs will be identified via comprehensive and systematic search of MEDLINE through PubMed, Google Scholar, Web of Sciences, Academic Search Premier and Complete (on EBSCO host), Karolinska Institutet Open Archive, Scopus, PsycINFO, Cochrane Library, SciELO, Centre for Reviews and Dissemination Databases (e.g., Database of Abstracts of Reviews of Effects [DARE]), International Prospective Register for Systematic Reviews (PROSPERO) register (for submitted protocols related to the specified health conditions and which may become available to summarize a topic area in the near future), Joanna Briggs Institute (JBI) Library of SRs and Implementation Reports, and Science Citation Index. A search of several databases is motivated by global focus of our research and intention to be as comprehensive as possible, especially considering our vulnerable or underrepresented in research study population. No language, year or publication status restrictions will be applied for the eligible studies during screening or study selection process. However, only “male sex” and “review” filters will be applied to avoid including articles about bisexual females and primary research, respectively.

The final search strategy has been decided by consensus of two reviewers and members of the Prevention and Implementation Group at the International Agency for Research on Cancer (IARC/WHO, Lyon, France), which include both information specialists and senior epidemiologists.

The list of key words related to cancers, their epidemiology, and MSM will include incidence, prevalence, survival, mortality, risk factors, and cancer surveillance, including registry. For the search engines that lack restriction options for SRs, alternative key words (e.g., “systematic review” OR “review”) will be used to identify SRs. A manual review of references for eligible SRs will also be performed. The example of search queries is provided in Table 3.

### 2.5. Data Extraction

Two reviewers will independently screen all articles identified from the search, following initial removal of duplicate and non-relevant materials. First, title and abstracts of the articles obtained from the initial search will be screened based on the aforementioned eligibility criteria. Second, full articles will be screened for their applicability. Finally, references of all considered articles will be hand-searched to identify any relevant publications that were not captured by the electronic search. Any disagreement between the two main reviewers will be resolved through discussion and if consensus is not possible, third and fourth reviewers will weigh in.

There may be an overlap of included studies in the identified systematic reviews, which will be apparent upon full text examination. When the overlaps occur, their frequency will be examined, and reported in an annex in full publication (e.g., using Venn diagram [25]). The most up-to-date available reference management software package (e.g., EndNote™) will be used and Excel tables will be created to help the research team manage potential overlap or duplication of SRs.

### 2.6. Data Collection

From each eligible SR, two reviewers will extract information independently on the following items: (i) full reference including first author, year of publication, number of included studies, population size, population characteristics (e.g., age and country of residence); (ii) anatomic site, histological and morphological characteristics, and related information on cancer; (iii) study design; (iv) setting or coverage (one-centre or multi-centric study); (v) lengths of study (e.g., follow up time); (vi) effect size (e.g., relative or absolute risks and corresponding 95% confidence intervals); and (vii) main independent variable(s), confounding variables or covariates, adjusted for in multivariable analyses, and effect modifiers. The reviewers will contact the authors of the publications with missing or unclear information on the aforementioned items.

### 2.7. Quality and Risk of Bias Assessment

A Measurement Tool to Assess Systematic Reviews (AMSTAR) will be used to assess the methodological quality and potential biases of the included SRs [32]. This instrument is made of 16 items having good content validity for measuring the methodological quality. Each item can receive a maximum of one point, for a possible range of AMSTAR scores of 0–11. The AMSTAR instrument will be administrated independently by two reviewers. Discrepancies or disagreements will be resolved via discussion with a third reviewer. Further sensitivity analyses (inclusion and exclusion) will be conducted on the studies with lower AMSTAR scores, and their impact will be summarized.

### 2.8. Methods for Evidence Synthesis

Quality of reporting will be appraised using the Preferred Reporting Items for Systematic Reviews and Meta-analysis (PRISMA) statement and the PRISMA checklist for reporting of SRs [33]. The data for each SR characteristics and findings based on methodological quality will be used to create UR tables. Extracted and summarized information will include specific details regarding the defined research questions, search strategy, eligibility criteria, studied population (e.g., sample size and participant characteristics) and included studies (e.g., type and number). The most recently published SRs of the highest quality (i.e., AMSTAR rating of overall confidence in the results of the review) and relevance to this UR purpose will be prioritized. For each exposure and outcome, tables or graphical displays will summarize characteristics of the included studies and their quality. The result and discussion sections of the SRs will be extracted to report their results and conclusions. Discussion sections will also be analyzed because they often contain further interpretation(s), which may offer important insights and enhance the richness of the findings in the current UR. Since one of the aims of an UR is to provide comparison of the effect sizes across exposures or risk factors, a common effect size will be estimated (e.g., by using approximate conversions). The UR will also report the heterogeneity across the studies and potential biases. Sensitivity analyses of reported findings will be considered, for example, by examining temporality of reported associations based on findings from prospective studies only [34]. To assure coherence of the review processes, 10% of the included reviews will be randomly selected, read and checked by both reviewers. They will critically examine accuracy of data extraction, including review characteristics (e.g., author, reference, aims and objectives, setting, number, and type of primary studies included). The quality of the evidence will be summarized using the Grades of Recommendations, Assessment, Development and Evaluation (GRADE) approach [35]. Finally, in case of missing data, reasons for missingness will be recorded if stated in the original SRs or provided by their authors through follow-up.

## 3. Conclusions

We foresee several limitations of the UR. First, because cancer initially emerged as a concern for MSM in the context of HIV/AIDS and surveillance of cancer-related behaviors in MSM population is emerging, we anticipate a predominant number of studies on the discovery and treatment of HIV/AIDS-related malignancies and the virology of cancer [22]. To supplement our search strategies inclusive of all cancers, we will look for studies on the aforementioned specific types of cancer in MSM. Second, challenges related to cancer surveillance in MSM (e.g., state of cancer surveillance and screening or early diagnosis programs, reliance on self-report of MSM status, as well as healthcare access and overall health status and quality of life of MSM) will translate into challenges with quantifying cancer outcomes. For example, the prevalence of prostate cancer in gay, bisexual, and other MSM ranging from 97,845 to 123,006 in the U.S., have been estimated by extrapolation [36]. Interpretation of the global estimates will also be context- or country-dependent given regional differences in data availability and quality, as well as state of health care systems [10].

Notwithstanding the above limitations, our UR will have several strengths. First, reliance on the AMSTAR and expert consensus will allow us to focus on methodological quality of systematic reviews and ensure our findings are based on high-quality reviews. Next, given the relative novelty of cancer research on MSM, a small number of systematic studies are expected, particularly from the geographic areas where MSM are stigmatized or discriminated. Furthermore, our UR will include SRs published in several languages, if available.

Because the UR will rely on retrieval and synthesis of data from SRs, approval by an ethics committee will not be required. Upon study completion and publication, the PROSPERO record will be updated. The findings will be disseminated using a variety of media outlets and audiences, including public health researchers and practitioners (e.g., through peer-reviewed publications, preferably, in open access journals, and conference presentations and workshops) and policy makers (e.g., through brief reports with summary of findings and recommendations). The UR will serve as an evidence-based decision-making tool for scientists, clinicians, and policy makers on cancer prevention and control strategies among MSM worldwide.

## Figures and Tables

**Table 1 ijerph-17-04954-t001:** Eligible Epidemiological Indicator and Definition.

Indicator	Definition
Incidence rate	The rate of new (or newly diagnosed) cases of the cancer. It is generally reported as the number of new cases of cancer occurring within a year per 100,000 population.
Prevalence	The actual number of cases; with the cancer either during a period of time (i.e., period prevalence) or at a particular date in time (i.e., point prevalence). Period prevalence provides a better measure of the cancer burden since it includes all new cases and all deaths between two dates, whereas point prevalence only counts those alive on a particular date.
Survival	The percentage of MSM population who are still alive for a certain period of time after they were diagnosed with or started treatment for cancer. The overall survival rate is often stated as a five-year survival rate, which is the percentage of MSM population in a study or treatment group who are alive five years after their diagnosis or the start of treatment for cancer.
Mortality rate	A cancer mortality rate is the number of deaths, with cancer as the underlying cause of death, occurring in the MSM population during a year. Cancer mortality is usually expressed as the number of deaths due to cancer per 100,000 of the MSM populations within a year.

Adapted from: SEER/NCI [30].

**Table 2 ijerph-17-04954-t002:** Adapted PECO (Populations, Exposure, Comparator, and Outcomes) Framework for Developing the Key Questions Examined in the Umbrella Review on Cancers among Men having Sex with Men (MSM) [31].

Element	Explanation
Population	Study population: all men who have sex with men including male bisexuals; disease: all cancers and specific types (e.g., anal, penile, prostate, liver, and lung cancers); setting: world-wide.
Risk factors and exposures	Compounds, exposure scenarios, as well as identified and potential risk factors for developing cancers, including health behaviors.
Comparator	The group to which population of interest is being compared: men who do not have sex with men.
Outcome	A deleterious change: overall and cancer-specific incidence rates, cancer and all-cause survival (e.g., 1-year or 5-year) and mortality rates, established and potential risk factors for all cancer types.

Adapted from Morgan et al. [31].

**Table 3 ijerph-17-04954-t003:** Potential Search Strategy Using Medline on PubMed.

Search	Query
#1	msm[tw] OR men who have sex with men[tw] OR ((“sexual minorities”[MeSH Terms] OR (“sexual”[All Fields] AND “minorities”[All Fields]) OR “sexual minorities”[All Fields] OR “homosexual”[All Fields] OR “homosexuality”[MeSH Terms] OR “homosexuality”[All Fields]) AND “male”[MeSH Terms]) OR gay[tw] OR bisexual[tw] AND (Review[ptyp] AND “male”[MeSH Terms])
#2	(“neoplasms”[MeSH Terms] OR cancer[Text word] OR tumor[Text word] OR tumour[Text word] OR malignancy[Text word]) AND (Review[ptyp] AND “male”[MeSH Terms])
#3	(epidemiology[Text Word] OR incidence[text word] OR prevalence[Text word] OR mortality[Text word] OR survival[Text Word]) AND (Review[ptyp] AND “male”[MeSH Terms])
#4	((registry[mesh] OR registry[text word])) AND Review[ptyp] AND Male[MeSH Terms]
#5	#1 AND #2 AND #3 AND (Review[ptyp] AND “male”[MeSH Terms])
#6	#1 AND #2 AND #4 (Review[ptyp] AND “male”[MeSH])

Notes: Other databases will be searched in same manner with slight modifications based on the characteristic of their respective search engine. To increase the search sensitivity, [text word] will be added to all search queries.

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
