# Peer review of "Epidemiology of Cancers in Men Who Have Sex with Men (MSM): A Protocol for Umbrella Review of Systematic Reviews"

_ijerph, 2020, doi:10.3390/ijerph17144954_

Round 1

Reviewer 1 Report

The topic of this manuscript is quite interesting to the readers of Int. J. Environ. Res. Public Health and the publication of the protocol of the study increase the reproducibility of the study. In addition, we do not understand why they authors want to publish this protocol if it has been published in PROSPERO. Moreover, there are some concerns that should include to improve the quality of the manuscript.

  1. In the introduction, the authors should be included a summarize the main results of the systematic review conducted about this issue as the aim defined better the justification of conducting this umbrella review. In addition, they should justify why is necessary to conduct an umbrella review to answer the aim of the study.
  2.  The authors should avoid the “consolidate” word in the aim of the study. The aim of this kind of the review is identify nature and extent of research evidence (usually including ongoing research) showed preliminary assessment of potential size and scope of available research literature.
  3.  In the methodology, the authors should include information about when they will performance the umbrella review. Moreover, more details about the eligibility criteria should be included. In addition, they should justify the need to us many secondary databases as BSCO host or Karolinska Institute Open Archive.
  4.  The authors should include in the methodology section information about the estimate a common effect size, heterogeneity, potential bias, stratification or sensibility analysis (you can see more detail of ten recommendations of the umbrella review in https://ebmh.bmj.com/content/21/3/95)
  5.  The conclusion should be eliminated of the protocol.

Author Response

Reviewer 1

The topic of this manuscript is quite interesting to the readers of Int. J. Environ. Res. Public Health and the publication of the protocol of the study increase the reproducibility of the study. In addition, we do not understand why they authors want to publish this protocol if it has been published in PROSPERO.

We apologize for confusion. The study protocol has been registered in PROSPERO rather than published, “To increase transparency, this study protocol has been registered with PROSPERO databases for systematic reviews and meta-analyses, a web-based international registry of protocols [CRD42017073377] of York University and National Institute for Health Research (NIH) (the United Kingdom).”

Moreover, there are some concerns that should include to improve the quality of the manuscript.

1. In the introduction, the authors should be included a summarize the main results of the systematic review conducted about this issue as the aim defined better the justification of conducting this umbrella review. In addition, they should justify why is necessary to conduct an umbrella review to answer the aim of the study.

Duly noted. In our response below, we tried to address this point and the next one.

2. The authors should avoid the “consolidate” word in the aim of the study. The aim of this kind of the review is identify nature and extent of research evidence (usually including ongoing research) showed preliminary assessment of potential size and scope of available research literature. 

Thank you. To address your comments 1 and 2, as well as the comment from the Academic Editor, we have modified our paragraph as follows, “Since several systematic reviews (SRs) on cancer epidemiology among MSM have been published [22-24], the goal of our research is to conduct an Umbrella Review (UR). While there are several definitions of umbrella reviews, their aim is to summarize available evidence, assess similarities and differences in published reviews, and perform different types of synthesis [25].  Hence, our UR will synthesize aspects of the current state of cancer surveillance on sexual orientation and knowledge on this unique population, including cancer risk factors, incidence, mortality, and survival compared to the non-MSM (i.e., population not included in the MSM definition). By doing so, the UR review will make substantive and methodological contributions to global epidemiology of malignancies among MSM. The purpose of this publication is to describe an UR protocol, which will guide our study and ensure its reproducibility.”

3. In the methodology, the authors should include information about when they will performance the umbrella review.

Thank you. We have added the following sentence, “We anticipate completing the UR in fall 2020.”

Moreover, more details about the eligibility criteria should be included. In addition, they should justify the need to us many secondary databases as BSCO host or Karolinska Institute Open Archive. 

In the “Study Population and Studies for Inclusion” subsection we have defined our study population and noted that “Only studies that have used a systematic process to search the literature and synthesize the data with standard reporting procedures (e.g., PRISMA) will be eligible for inclusion in the UR. It will not include abstracts, critical reviews, integrative reviews, primary studies, withdrawn/retracted publications, dissertations, and literature reviews.”

We have also clarified in the “Search Strategy” subsection that “Search of several databases is motivated by global focus of our research and intention to be as comprehensive as possible, especially considering our vulnerable or underrepresented in research study population.”

4. The authors should include in the methodology section information about the estimate a common effect size, heterogeneity, potential bias, stratification or sensibility analysis (you can see more detail of ten recommendations of the umbrella review in https://ebmh.bmj.com/content/21/3/95) 

Thank you for the resource by Fusar-Poli and Radua. We have added it to our reference list and the following sentences, “Since one of the aims of an UR is to provide comparison of the effect sizes across exposures or risk factors, a common effect size will be estimated (e.g., by using approximate conversions). The UR will also report the heterogeneity across the studies and potential biases. Sensitivity analyses of reported findings will be considered, for example, by examining temporality of reported associations based on findings from prospective studies only [34].”

5. The conclusion should be eliminated of the protocol.

Based in the comments from the Academic Editor, we kept this section with modifications addressing concerns of the Academic Editor and Reviewer 2. We would gladly make further modifications if needed upon further review.

Reviewer 2 Report

Dear writers,

The incidence of cancer in minorities and stigmatised groups in society is potentially under-investigated.

1) Cancer is a clinical condition characterised by uncontrolled cellular division, impaired cell cycle control, invasive growth, immune evasion and metastases.

2) It can be very difficult to quantify cancer incidence and prevalence as this is dependent on the overall health and life expectancy of the cohort of patients.

3) The literature search, review while interesting will have limitations as MSM are self reported.

4) Cancer rates (incidence and prevalence) also depends on the availability of screening programmes in order to diagnose early/ pre cancerous lesions. This will be biased to higher rates of cancer incidences in nations with advanced health care systems. (less stigma, more screening programmes)

Otherwise, methodology wise seems reasonable.

Thank you for your work.

Author Response

The incidence of cancer in minorities and stigmatised groups in society is potentially under-investigated.

1) Cancer is a clinical condition characterised by uncontrolled cellular division, impaired cell cycle control, invasive growth, immune evasion and metastases.

2) It can be very difficult to quantify cancer incidence and prevalence as this is dependent on the overall health and life expectancy of the cohort of patients.

3) The literature search, review while interesting will have limitations as MSM are self reported.

4) Cancer rates (incidence and prevalence) also depends on the availability of screening programmes in order to diagnose early/ pre cancerous lesions. This will be biased to higher rates of cancer incidences in nations with advanced health care systems. (less stigma, more screening programmes)

Thank you for your insight. We have added a new paragraph, which included all of your comments, “We foresee several limitations of the UR. First, because cancer initially emerged as a concern for MSM in the context of HIV/AIDS and surveillance of cancer-related behaviors in MSM population is emerging, we anticipate a predominant number of studies on the discovery and treatment of HIV/AIDS-related malignancies and the virology of cancer [22]. To supplement our search strategies inclusive of all cancers, we will look for studies on the aforementioned specific types of cancer in MSM. Second, challenges related to cancer surveillance in MSM (e.g., state of cancer surveillance and screening or early diagnosis programs, reliance on self-report of MSM status, as well as healthcare access and overall health status and quality of life of MSM) will translate into challenges with quantifying cancer outcomes. For example, the prevalence of prostate cancer in gay, bisexual, and other MSM ranging from 97,845 to 123,006 in the U.S., have been estimated by extrapolation.[36] Interpretation of the global estimates will also be context- or country-dependent given regional differences in data availability and quality, as well as state of health care systems.[10]”

Reviewer 3 Report

The authors present comprehensive guidelines how to perform epidemiological studies on cancers in MSM population.

What is the relation of authors to IARC? Is the protocol an official document of IARC or not?

Line 60: format reference 12 according to journal format.

Tables 1 and 2 should be formatted so that it is better readable with empty lines inbetween or with frames.

Author Response

The authors present comprehensive guidelines how to perform epidemiological studies on cancers in MSM population.

What is the relation of authors to IARC? Is the protocol an official document of IARC or not?

Three of the authors were employed at IARC at the time of writing this manuscript. The protocol will provide foundation for potential future IACR documentation, but it will not be its official document. 

Line 60: format reference 12 according to journal format.

Thank you. We have highlighted the year in reference 12 as required by the journal, and added a period after the journal name. We did the same for the rest of the references.

Tables 1 and 2 should be formatted so that it is better readable with empty lines inbetween or with frames.

We have reformatted Tables 1 and 2.

Round 2

Reviewer 1 Report

I appreciate the effort in addressing the prior comments and you hard work in this new version of the manuscript. In my opinion, the aim of the publication of the protocol of the study is to increase the reproducibility of the study. In fact, this is the main function of PROSPERO register.

Author Response

Thank you. We concur. We have made sure the final sentence in the Introduction section reads as follows, "The purpose of this publication is to describe an UR protocol, which will guide our study and ensure its reproducibility."

Very respectfully.

Vitaly Smelov, MD, PhD